# Survival in People Living with HIV with or without Recurrence of Hepatocellular Carcinoma after Invasive Therapy

**DOI:** 10.3390/cancers15061653

**Published:** 2023-03-08

**Authors:** Costanza Bertoni, Laura Galli, Riccardo Lolatto, Hamid Hasson, Alessia Siribelli, Emanuela Messina, Antonella Castagna, Caterina Uberti Foppa, Giulia Morsica

**Affiliations:** 1Division of Infectious Diseases, IRCCS San Raffaele Scientific Institute, 20132 Milan, Italy; 2Faculty of Medicine and Surgery, Vita-Salute San Raffaele University, 20132 Milan, Italy

**Keywords:** hepatocellular carcinoma, people living with HIV, recurrence, invasive therapy, survival

## Abstract

**Simple Summary:**

Few data are available on HCC outcomes in people living with HIV (PLWH), especially regarding HCC treatment and recurrence. We focused our study on survival and recurCitation: Bertoni, C.; Galli, L.; Lolatto, R.; Hasson, H.; Siribelli, A.; Messina, E.; Castagna, A.; Uberti-Foppa, C.; Morsica, G. rence in HCC/PLWH under invasive therapy by considering also associated factors. To the best of our knowledge, this is the first study investigating survival and recurrence in HCC/HIV people who received invasive therapy, showing that HCC/PLWH under invasive therapy can achieve a good 2- and 5-year survival regardless of HCC recurrence or not. We also showed that the best outcome in terms of survival was obtained in transplanted participants. Our findings, although obtained in a small sample size, suggest that HCC/PLWH should have the same treatment opportunities as HIV-negative participants also in terms of re-treatment. As for the counterpart of HIV-uninfected participants, a more aggressive treatment than what was recommended by the BCLC system could be offered after careful selection to HCC/PLWH.

**Abstract:**

Background and Aims: To address the overall survival (OS) and recurrence (RE) in people living with HIV (PLWH) treated with invasive therapy (IT) for hepatocellular carcinoma (HCC). Methods: This is a retrospective cohort study on 41 PLWH with HCC receiving IT, defined as liver resection (LR), orthotopic liver transplantation (OLT), radiofrequency thermo-ablation (RFTA) trans arterial chemo, or radioembolization (CRE). OS and RE were investigated by Kaplan–Meier curves. The Cox proportional hazard regression model was used for multivariate analyses. Results: Recurrence occurred in 46.3% PLWH; in 36.7% of participants at 2 years and in 52% at 5 years from HCC diagnosis; it was less frequent in males, *p* = 0.036. Overall, 2- and 5-year survival after HCC diagnosis was 72% and 48%, respectively. Two-and five-year survival was 100% and 90.9%, respectively, in PLWH receiving OLT, compared to other IT (60.9% and 30.6%, respectively) log-rank *p* = 0.0006. Two- and five-year survival in participants with no-RE was 70.5% and 54.6%, respectively, and 73.7% and 42.1% among RE, respectively, log-rank *p* = 0.7772. By multivariate analysis, AFP at values < 28.8 ng/mL, at HCC diagnosis, was the only factor predicting survival. Conclusions: Fifty percent of PLWH survived five years after HCC diagnosis; 90.9% among OLT patients. Recurrence after IT was observed in 46% of HCC/PLWH. AFP cut-off levels of 28.8 ng/mL were the only independent variable associated with survival.

## 1. Introduction

Improvements in the care of PLWH, including potent antiretroviral therapy (ART), have resulted in a decline in HIV-related deaths; so, the life expectancy of HIV individuals has increased [1,2]. Chronic liver diseases and end-stage liver diseases, which are primarily related to complications of chronic hepatitis B virus (HBV) or hepatitis C virus (HCV) coinfection, as well as hepatotoxicity exerted by ART, now account for up to 50% of deaths among PLWH [3,4,5]. Multiple studies show that PLWH are 6–7 times more likely to develop HCC than those who are not infected with HIV; this is commonly attributed to the high prevalence of coinfection with viral hepatitis and their immune-suppressed status [5,6,7].

Opinions regarding the survival status of PLWH who underwent anticancer treatment differ. Some reported shorter survival for PLWH with HCC, irrespective of tumor stage [7,8,9], while others showed a similar survival rate when care management was standardized [10,11].

Studies showed that radical treatment, such as surgical resection, is more likely for PLWH due to the earlier stage of the tumor and preserved liver function [7], which is considerably different from the results of other studies [8,12]. However, the participants included in these previous studies underwent different therapeutic strategies, including invasive curative therapy, non-curative therapy, or best supportive care, according to the condition of HCC and functional reserve of the liver.

Despite the fact that current therapeutic strategies have significantly improved mortality rates over the last decades, the prognosis of HCC is generally poor, and recurrence will happen after treatment, leading to treatment failure [13,14].

To date, studies have rarely investigated the presentation of participants who underwent invasive therapy (IT) for an early or intermediate HCC stage, and few data are available on long-term survival status and recurrence in HCC/PLWH according to invasive therapy. 

Classic predictors for the prognosis of HCC and risk of recurrence focus on tumor burden, cancer-related symptoms, and other common features such as tumor size and multinodularity. The 2022 BCLC model [15] included parameters related to liver function such as albumin-bilirubin score (ALBI) and biomarkers (alpha-fetoprotein, AFP), possibly reflecting HCC malignancy and its ability to metastasize.

Previous studies have indicated the systemic inflammatory markers, such as the neutrophil-to-lymphocyte ratio (NLR) [16,17], platelet-to-lymphocyte ratio (PLR) [17], C-reactive protein-to-albumin ratio (CAR) [18], as predictors of HCC outcome. However, these markers may not reflect details about immune status. Recently, some studies have explored the circulating immune perturbations that occurred in patients with HCC as predictors of disease and therapeutic outcomes [19,20].

Circulating CD4 cells usually decrease in the progressive stages of HCC [21]. Additionally, PLWH have low peripheral blood CD4 cell counts. Assessment of the peripheral immune status in PLWH could provide prognostic information and further aid, together with other clinic parameters such as AFP concentration, ALBI score, and inflammatory markers, for personalized treatment options and possibly early recognition of recurrence. Therefore, we decided to evaluate clinical prognostic predictors, survival, and recurrence of HCC in PLWH who received invasive therapeutic approaches.

## 2. Materials and Methods

### 2.1. Study Population

This is a retrospective study on 41 adult PLWH with incident HCC diagnosed between November 2000 and October 2021, treated with IT defined as liver resection (LR), orthotopic liver transplantation (OLT), radiofrequency thermo-ablation (RFTA), trans arterial chemo, or radioembolization (CRE). Data were collected as part of routine clinical care and recorded in the database of the Division of Infectious Diseases of the San Raffaele Hospital (CSLHIV Cohort). The CSLHIV Cohort was approved by the Ethics Committee of the San Raffaele Hospital, Milan, Italy.

On their first visit, PLWH provided written informed consent for the use of their data in scientific analyses. Recorded data were anonymized and managed according to good clinical practice. Information on demographics, mortality, and biochemistry measured at baseline (BL), defined as the time of HCC diagnosis, at IT, and at the last available evaluation (LE), were considered in the present study: serum levels of alfa-fetoprotein (AFP), laboratory values concerning liver function [aspartate aminotransferase (AST) alanine aminotransferase (ALT), total bilirubin, prothrombin time in seconds (PTs), international normalized ratio (INR) platelets count, and HIV parameters. 

Alpha-fetoprotein levels were also stratified according to the cut-off levels of 28 ng/mL, which we previously found associated with survival in PLWH with HCC [22]. Hepatitis B, C, and Delta infections were assessed by the presence of hepatitis B surface antigen [HBsAg], hepatitis C, and hepatitis Delta antibody positivity (anti-HCV and anti-HDV, respectively).

Participants included in the study were diagnosed as having liver cirrhosis according to the following criteria: (a) biopsy proven, (b) esophageal varices with endoscopic documentation, and (c) thrombocytopenia (≤100 × 10^9^/L) with ultra-sound (US) signs of portal hypertension.

HCC diagnosis was based on imaging or histologic criteria according to international guidelines [23]. Portal vein thrombosis [PVT] was detected using US with doppler examination. The PLWH were staged according to the Child–Turcotte–Pugh (CTP) functional class and Barcelona Clinic Liver Cancer (BCLC) algorithm, following pre-published methodology [24,25,26].

### 2.2. Statistical Analysis

Participants’ characteristics were described as median (interquartile range, IQR) for continuous variables or proportions for categorical variables. Continuous variables were compared using the Wilcoxon rank sum test, while differences between proportions were tested by the chi-square or Fisher’s exact test.

Kaplan–Meier curves on survival or recurrence were calculated and compared by use of the log-rank test. Time to death or recurrence since HCC diagnosis (baseline, BL) was censored at the date of death or recurrence or lost to follow-up or data freezing on February 2022 (whichever occurred first).

A multivariable Cox regression model was assessed to explore factors associated with the risk of survival. Variables known to have a potential effect on this outcome were considered to obtain the multivariable model and included: sex, baseline AFP, CD8 cell count, and total lymphocytes; the number of variables was limited and based on the number of events in the sample (we considered five events per predictor as a rule) [27]. The adjusted hazard ratios (aHR) of survival were reported with the corresponding 95% CI for significant covariates.

A two-sided *p*-value < 0.05 was considered to be statistically significant. All analyses were conducted using SAS statistical software version 9.4 (Statistical Analyses System Inc., Cary, NC, USA).

## 3. Results

### 3.1. Baseline Characteristics of HCC/PLWH with a Fatal Outcome or Not

Baseline clinical characteristics of 41 HCC/PLWH who received invasive treatment according to the outcome survival are summarized in Table 1.

Overall, there was a preponderance of males (83%), and intravenous drug use (IVDU) was the more prevalent risk factor for HIV infection (53.7%). Regarding characteristics of the tumor, most participants (93%) had <3 nodules; portal vein thrombosis and extra-hepatic spread of the tumor were observed in 29.3% and 9.8% of PLWH, respectively.

Concerning HCC etiology, 32 participants (78%) were infected by HCV, 8 (19.5%) by HBV, and 1 patient had a double HBV/hepatitis D virus [HDV] infection. These 9 participants with HBV or HBV/HDV infection were under treatment with drugs active against HIV/HBV: tenofovir disoproxil fumarate and emtricitabine. Among HCV-infected participants, HCV genotype (GT) was available in 23 participants: HCV GT3 was the most prevalent (12/23, 52.2%); the other genotypes were GT1a in 7, GT4 in 3 other participants, and GT1b in the remaining one participant. Fifteen of the HCV-infected participants were treated with direct-acting antivirals, all of whom reached viral eradication, while five other participants with HCV received interferon alpha associated with ribavirin treatment, with viral eradication in three of them and no response to treatment in the other two participants.

A comparison of demographics at BL between died and alive participants showed that male sex was more frequent (95%) among alive than among died participants, *p* = 0.01. The factors attaining HCC characteristics showed no differences between died and alive participants with respect to the distribution of portal vein tumor thrombus and extra-hepatic metastasis, while PLWH with a fatal outcome had more frequently a multinodular disease, *p* = 0.01. Regarding biochemistry, a higher BL AFP was found in died participants compared to alive people: median levels, 41.4 (interquartile range, IQR: 14.6–347.7) vs. 14.4 (IQR: 7.3–24.0) ng/mL, *p* = 0.015.

Considering the baseline AFP cut-off level of 28 ng/mL, 6/19 [31.5%] individuals with AFP < 28 ng/mL had a fatal outcome, compared to 15 [83.3%] of those with an AFP ≥ 28 ng/mL, *p* = 0.002. A lower CD8 cells count, median 589 cells/mmc (IQR: 307–757) vs. median 1021 cells/mmc (IQR: 621–1392) *p* = 0.02; and a lower but not significantly different total lymphocyte count, median 1.5 × 10^9^/L (IQR 1–2.2) vs. median 2.3 × 10^9^/L (IQR 1.6–2.6) *p* = 0.05 and neutrophils count, median 2.6 (1.8–3.2) vs. 3.3 (2.4–4.4) *p* = 0.082, were also detected in people with a fatal outcome compared to those with a favorable outcome. The other variables were similarly distributed in the two groups of HCC/PLWH.

Of 41 HCC/PLWH, 6 (14.6%) received liver resection [LR]; 11 (26.8%) orthotopic liver transplantation (OLT); 6 (14.6%) radiofrequency thermo-ablation (RFTA); and 18 (43.9%) chemo/radioembolization (CRE).

OLT recipients had the best prognosis, while people who underwent CRE had the worst prognosis, *p* = 0.0038 (Table 1).

### 3.2. Biochemistry at Invasive Therapy

The participants underwent IT after a median time of 57 days since HCC diagnosis (IQR: 27–218). Biochemistry at IT is summarized in Table 2. Higher values of alpha fetoprotein, *p* = 0.036, aspartate aminotransferase (AST) levels, *p* = 0.03, total bilirubin, *p* = 0.005, and prothrombin time in seconds/international normalized ratio, PTs/INR, *p* = 0.008, were observed among died participants compared to alive people, while CD4 cells count, *p* = 0.035, CD8 cells count, *p* = 0.0007, total lymphocytes, *p* = 0.005, pseudocholinesterase, (PCHE) *p* = 0.001, and creatinine, *p* = 0.005, were significantly lower in participants who died with respect to alive people.

### 3.3. Biochemistry at Last Evaluation

Biochemistry at the last available evaluation is summarized in Table 3. After a median follow-up of 4.03 years (IQR: 1.14–7.07), higher median AFP levels, *p* < 0.0001 as well as AFP values according to the cut-off levels of 28 ng/mL, *p* = 0.0002, were observed in HCC/PLWH who died. Albumin, *p* = 0.002, PCHE, *p* = 0.0002, but not PTs/INR and total bilirubin were worse in died participants, with also evidence of a decrease in CD4, *p* = 0.0004, CD8, *p* = 0.0005 and total lymphocytes, *p* = 0.002.

We also found a higher neutrophils/lymphocytes ratio, *p* = 0.001, and necro-inflammatory activity assessed by AST and ALT levels, *p* = 0.0001 and *p* = 0.008, respectively, in participants with a fatal outcome compared to those with a favorable outcome. Finally, creatinine levels were found lower in HCC/PLWH who died compared to alive participants, *p* = 0.023.

### 3.4. Characteristics of HCC/PLWH and Recurrence

After invasive therapy for HCC, recurrence was observed in 19/41 (46.3%) PLWH [median time to recurrence: 1.34 years (IQR: 0.56–3.04)]: in 4/11 (36.4%) participants who received OLT as first therapy, in 4/6 (66.7%) with LR, in 4/6 (66.7%) of those who received RFTA, in 7/18 (38.9%) of participants who underwent CRE, *p* = 0.418. Overall, the 2- and 5-year probability of recurrence was 36.7% (IQR: 23.1–55.0%) and 52% (IQR: 35.7–70.6%), respectively (Figure 1).

Characteristics of PLWH evaluated at HCC diagnosis, at IT, and at the last visit available after IT, according to the event of recurrence, are described in the Appendix A (online Appendix A). At BL evaluation, the only variable significantly associated with recurrence was the male sex [RE 13 (68.4%) vs. no-RE 21 (95.5%) *p* = 0.036]; all the other variables were similarly distributed.

At invasive therapy, neutrophils, median numbers 2.95 cells × 10^9^/L (IQR: 2.3–4.2) vs. 4.55 × 10^9^/L (IQR: 2.9–5.4) *p* = 0.04; and creatinine levels [median value 0.82 mg/dL (IQR: 0.71–1.08) vs. median value 1.01 mg/dL (IQR: 0.83–1.48) *p* = 0.018] were differently distributed between participants with RE and those with no-RE, while at LE all the variables examined were similarly distributed.

### 3.5. Survival

Overall, 22/41 (54%) HCC/PLWH treated with IT died during a median follow-up of 4.03 years (IQR: 1.14–7.07). The 2- and 5-year survival probabilities were 72% (IQR: 55.1%–83.4%) and 48% (IQR: 31.7%–62.7%), respectively (Figure 2).

The median follow-up was higher in OLT respect to participants with other treatments (LR, RFTA, CRE) [8.58 years (IQR: 6.02–9.99) vs. 2.46 years (IQR: 1.02–5.35)] and associated with a higher rate of survival: the 2- and 5-year survival was 100% and 90.9%, respectively, in OLT in comparison with other therapies (LR, RFTA, and CRE) (61.0% and 30.6%, respectively), log-rank *p* = 0.0006 (Figure 3).

Two- and five-year survival according to RE was 73.7% (IQR: 47.9–88.1%) and 42.1% (IQR: 20.4–62.5%), respectively, in PLWH with RE and 70.5% (IQR: 45.7–85.6%) and 54.6% (IQR: 30.6–73.4%), respectively, in PLWH with no-RE, log-rank *p* = 0.7772 (online Appendix A).

By multivariate analysis, AFP at the value of <28 ng/mL evaluated at HCC diagnosis, was the only factor associated with a better prognosis (adjusted hazard ratio, aHR = 0.271, 95% confidence interval, CI = 0.078–0.948, *p* = 0.041), after adjusting for male sex, CD8 cell count, and total lymphocytes.

## 4. Discussion

According to the available epidemiological data, deaths related to chronic liver disease complications, mainly HCC, have significantly increased in recent years in Western countries, suggesting that HCC must be considered an important emerging cause of death in PLWH [28,29].

Few data are available on HCC outcomes in PLWH, especially regarding HCC treatment and recurrence. Therefore, we focused our study on survival and recurrence in HCC/PLWH under invasive therapy by considering also associated factors.

Previous studies highlighted the interplay between the liver reserve and immune dysfunction as a potential prognostic factor for the survival of HCC/PLWH [30,31]. At BL evaluation, most participants had CD4 > 200 cells/mmc, and only 4 participants had a detectable HIV load, indicating a relatively preserved immune status and good control of HIV viremia by antiretroviral therapy. Interestingly, we found an initial perturbation of immune status (BL evaluation) which involved lymphocytes and CD8 cells in participants with a fatal outcome, followed at subsequent time points (IT and LE) by a concomitant decrease in CD4 cell count. This finding suggests an increasing severity of the underlying immune disturbance during the period of observation, possibly because of the progression of liver dysfunction in HCC/PLWH with a fatal outcome. In fact, the biochemistry reflecting liver functional reserve (i.e., albumin, bilirubin, PCHE, and PT/INR) progressively worsened in PLWH who died compared to alive participants (see Table 1, Table 2 and Table 3). The characteristics of the tumor (multinodular disease and high AFP levels), possibly reflecting HCC malignancy and its ability to metastasize, seemed related to a worse prognosis. This is in line with previous reports [32,33,34,35,36] that identified characteristics of tumors and AFP levels as independent prognostic factors. Additionally, the type of treatment but not CPT classification was associated with a favorable outcome, while the BCLC score showed a trend toward significance. 

The male sex was less frequent in recipients with recurrence. This is a surprising finding because it is well known that men have a greater risk of developing HCC than women worldwide, and female sex is considered a favorable prognostic factor due to a higher survival rate and features associated with early disease [37]. Reports have suggested that estrogen acts as an inhibitor of the proliferation, growth, and metastasis of HCC cells and can prevent liver cancer development [38]. Several studies have suggested that the androgen receptor (AR), but not androgen, could be an important target of hepatocarcinogenesis and HCC development [39,40]. Other studies have shown that AR may inhibit the metastasis of late-stage HCC [41]. It was also reported that AR has an inhibitory role in the carcinogenesis of HCC by lowering the recurrence rate in post-surgery patients [42]. Although we did not investigate the androgen pathway, we can hypothesize that androgens can function in a contrasting manner, as can the estrogen pathway, in the context of recurrence after invasive therapy. 

Numerous studies on HIV-negative participants have identified the utility of elevated AFP in predicting HCC recurrence after initial hepatic resection or other invasive therapy [43,44,45,46] while other investigators have disputed its prognostic role for HCC recurrence [47,48]. To the best of our knowledge, no data on this topic are currently available in HCC/PLWH. In this study, AFP protein at our optimal estimate of 28.8 ng/mL, as well as median levels, seemed not to be associated with a higher probability of recurrence. However, several participants in our study underwent chemo/radioembolization, while these previous reports [43,44,45,46] investigated the effect of AFP levels after a more aggressive therapy used for an earlier of stage HCC.

Concerning survival, clinical studies have demonstrated that HCC/PLWH can obtain a satisfactory survival rate after treatment [6,8,49]. Although 18 (43.9%) PLWH received non-curative IT (chemo/radioembolization) as their first therapy, we had a good overall survival because approximately 50% of our participants were alive after 5 years from their HCC diagnosis. Additionally, 32% of PLWH were on BCLC C/D, and the 5-year survival was 31% compared to 54% of participants classified with an A/B BCLC score, although not statistically significant (*p* = 0.2). We have a longer survival duration with respect to other studies performed in PLWH [6,9,50,51]. Notably, except for the study by Zhao et al. [51], in which there were enrolled participants who underwent LR, these reports have a different design because the participants were subjected to diverse therapeutic strategies, including curative therapy, palliative therapy, or no therapy according to the condition of HCC.

Regarding transplantation, the two- and five-year survival rates were 100% and 90.9%, respectively, in eleven OLT recipients, despite the fact that four of them had tumor recurrence. According to other studies [52,53,54,55], our data suggest that PLWH do derive a significant survival benefit from liver transplantation.

It is well known that tumor recurrence is still the main factor contributing to the poor prognosis of HCC patients, especially under non-curative treatment such as chemo/radioembolization. To the best of our knowledge, only one study has investigated survival according to the outcome of recurrence in HCC/PLWH [6]. They showed 70.7% recurrence and 10.3% 5-year survival. 

We had a 46.3% recurrence rate and a higher rate of survival according to whether there was a recurrence or not (48% and 53.6%, respectively) after 5 years from the HCC diagnosis. The more favorable outcome in our group of HCC/PLWH is probably due to the different design of the study since we included exclusively people who received IT, compared to the report by Berretta et al. [6], in which patients receiving different therapies were considered too: Patients under systemic therapy (24.7% of PLWH) or under any therapy that affects survival rate because of poorer liver function and/or an inoperable tumor. However, similarly to the study by Berretta et al. [6], we showed that PLWH had recurrence after a median period of 1.34 years.

In clinical practice, a right-to-left migration strategy (i.e., a therapy recommended for earlier stages) is not infrequent within the BCLC algorithm after careful selection of participants. In this study, several participants received a more aggressive treatment than what was recommended by the BCLC system; in particular, 7 participants in BCLC stage C and 3 in BCLC stage D received CRE, and one HIV/HBV-coinfected patient in BCLC stage D and CTP C was transplanted. Their 5-year survival was lower (31%), but not significantly different from that of participants who had an early or intermediate HCC stage and a more preserved liver function (54% 5-year survival).

Very recently, we showed in PLWH with HCC that AFP at a precise cut-off is associated with a worse prognosis, considering all treatment options [22]. In the present study, which included PLWH who received IT, the AFP at values < 28 ng/mL emerged as an independent survival-related factor, confirming the importance of this biological marker in the prognosis of HCC also at an early stage since 48% of participants had BCLC 0/A.

The current study had several limitations that should be taken into consideration when interpreting the results. The retrospective nature of the study is potentially connected with the following limitations: Several HCC/PLWH were followed during years in which HCC treatment options available for HIV-negative participants were not approved in PLWH; the surveillance tests for HCC could not be performed at optimal intervals in some participants with HCC diagnosis in a more advanced stage; antiretroviral medication, as well as anti-HCV therapy, have greatly evolved in recent years with increased efficacy and better tolerability, lowering the risk of progression of liver diseases, also in the context of HCC.

A second limitation is that the study was conducted in a single center; therefore, generalizations of our findings may be limited. However, in our hospital, the care plan of HCC participants is discussed at the multidisciplinary tumor board. Therefore, treatment options offered to participants for recurrence are more homogeneous with respect to multi-cohort studies.

Third, the sample size available for our analyses was small and therefore associated with low statistical power in detecting differences among died and alive participants and with low accuracy of reported estimates. Our study, on the other hand, can be enumerated among the few studies in the field of survival and recurrence involving HCC/PLWH.

## 5. Conclusions

In conclusion, our study suggests that HCC/PLWH under IT can achieve a good 2- and 5-year survival regardless of HCC recurrence or not. The best outcome in terms of survival was obtained in transplanted participants. The independent factor predicting survival in our group of HCC/PLWH was AFP at a precise cut-off.

These findings also suggest that HCC/PLWH should have the same treatment opportunities as HIV-negative participants in terms of re-treatment. As for the counterpart of HIV-uninfected participants, a more aggressive treatment than what was recommended by the BCLC system could be offered after careful selection to HCC/PLWH. Future and more extensive studies are warranted to focus on the stratification of PLWH at high risk for HCC progression who would benefit from a specific invasive treatment.

## Figures and Tables

**Figure 1 cancers-15-01653-f001:**
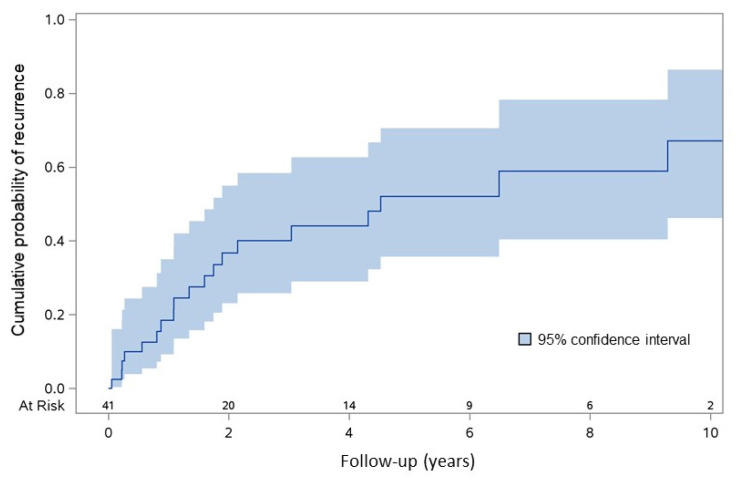
Cumulative probability of recurrence in HCC/PLWH who received IT.

**Figure 2 cancers-15-01653-f002:**
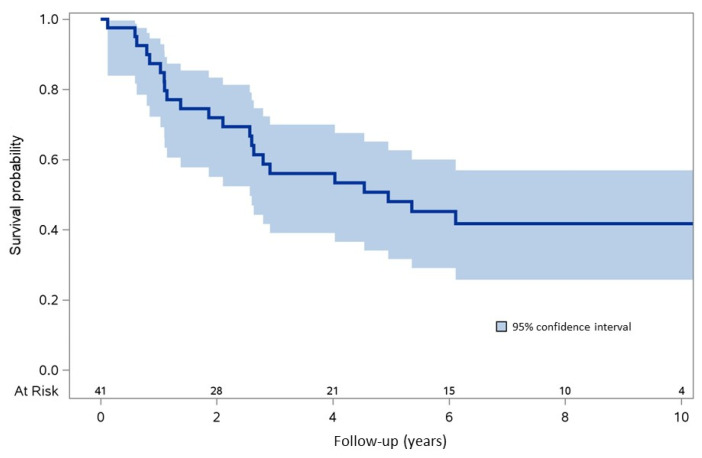
Overall survival in HCC/PLWH who received IT.

**Figure 3 cancers-15-01653-f003:**
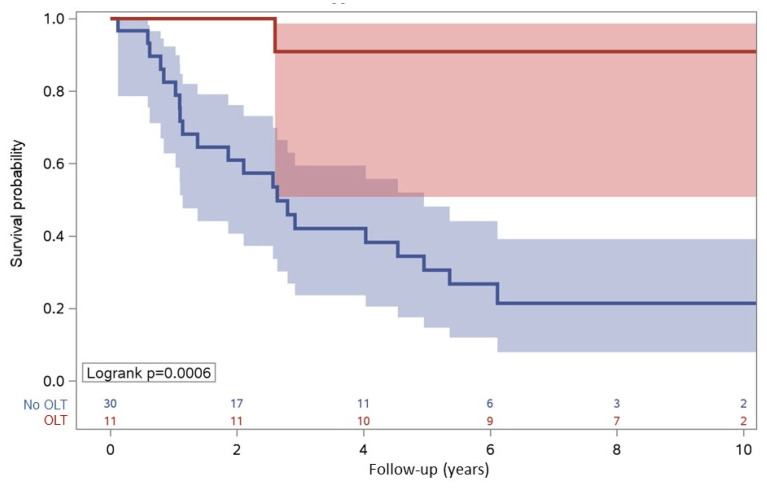
Survival according to OLT or other invasive therapy including LR, RFTA, CRE.

**Table 1 cancers-15-01653-t001:** Characteristics at HCC diagnosis of PLWH receiving invasive therapy according to the primary study outcome (died or alive).

Variable		Overall (Number = 41)	Died (Number = 22)	Alive (Number = 19)	*p*-Value
Age, years		53 (49–56)	53 (49–56)	54 (50–58)	0.267
Sex, male		34 (83)	16 (72)	18 (95)	0.010
Years of HIV infection		22.75 (12.67–26.41)	21.02 (10.11–25.34)	22.75 (16.22–27.08)	0.278
Years since first ART		11.95 (7.29–16.84)	10.64 (5.11–15.9)	14.95 (8.78–19.39)	0.206
Number of nodules > 3		3 (7)	3 (14)	0 (0)	0.010
Cancer embolus in portal vein	No	29 (70.7)	15 (68.2)	14 (73.7)	0.744
	Yes	12 (29.3)	7 (31.8)	5 (26.3)	
Extra-hepatic spread	No	37 (90.2)	19 (86.4)	18 (94.7)	0.610
	Yes	4 (9.8)	3 (13.6)	1 (5.3)	
Child Pugh Turcotte	A	33 (80.5)	17 (77.3)	16 (84.2)	0.563
	B	7 (17.1)	4 (18.2)	3 (15.8)	
	C	1 (2.4)	1 (4.5)	0	
BCLC	0/A	20 (48.8)	7 (31.8)	13 (68.4)	0.063
	B	8 (19.5)	6 (27.3)	2 (10.5)	
	C/D	13 (31.7)	9 (40.9)	4 (21.1)	
Treatment					0.0038
	OLT	11 (26.8)	1 (4.5)	10 (52.6)	
	LR	6 (14.6)	4 (18.2)	2 (10.5)	
	RFTA	6 (14.6)	3 (13.6)	3 (15.8)	
	CRE	18 (43.9)	14 (63.6)	4 (21.0)	
AFP, ng/mL		27.8 (8.7–135.2)	41.4 (14.6–347.7)	14.4 (7.3–23.95)	0.015
AFP, ng/mL	<28.8	19 (51.4)	6 (28.6)	13 (81.3)	0.002
	≥28.8	18 (48.6)	15 (71.4)	3 (18.8)	
AST, U/L		70 (34–100)	71 (38–100)	45 (31–158)	0.855
ALT, U/L		67 (33–99)	64 (32–99)	68 (35–104)	0.466
Bilirubin, mg/dL		1 (0.78–1.75)	1 (0.78–1.52)	1 (0.78–1.75)	0.824
Albumin (g/L)		39.9 (37.26–41.11)	39.9 (37.26–41.11)	40.31 (33.78–43.03)	1.000
CD4 cells count, mmc		433 (262–722)	333.5 (246–722)	530 (345–764)	0.218
CD8 cells count, mmc		732 (445–1209)	589 (307–757)	1021 (621–1392)	0.02
CD4/CD8 ratio		0.68 (0.34–0.98)	0.71 (0.35–1.11)	0.42 (0.31–0.94)	0.264
Neutrophils, cells 10^9^/L		2.85 (2.4–3.4)	2.6 (1.8–3.2)	3.3 (2.4–4.4)	0.082
Lymphocytes, cells 10^9^/L		1.7 (1.2–2.6)	1.5 (1–2.2)	2.3 (1.6–2.6)	0.05
Neutrophils/lymphocytes, ratio		1.58 (1.12–2.14)	1.36 (1.06–2.36)	1.59 (1.28–2.11)	0.521
Platelets count, 10^9^/L		107 (64–141)	100 (64–123)	110.5 (63–154)	0.325
PTs/INR		1.06 (1.03–1.28)	1.06 (0.99–1.37)	1.08 (1.05–1.21)	0.953
Creatinine, mg/dL		0.74 (0.68–0.86)	0.72 (0.66–0.88)	0.75 (0.72–0.86)	0.444
Anti-HCV positive		32 (78)	17 (77.3)	15 (78.9)	0.976
HBsAg positive		8 (19.5)	4 (18.2)	4 (21.1)	0.883
HIV-RNA > 50 copies/mL		4 (9.8)	2 (9.1)	2 (10.5)	0.560
	Unknown	14 (34.1)	6 (27.3)	8 (42.1)	

Results are described by median (IQR) or frequency (%). Abbreviations: ART: antiretroviral therapy; AFP: alpha phetoprotein; PTs/INR: prothrombin in second/international normalized ratio; HCV: hepatitis C virus; HBsAg: hepatitis B surface antigen; AST: aspartate aminotransferase (normal values < 35 U/L); ALT: alanine aminotransferase (normal values < 59 U/L); BCLC: Barcelona Clinic Liver Cancer.

**Table 2 cancers-15-01653-t002:** Biochemistry of PLWH at invasive therapy for HCC according to the primary study outcome (died or alive).

Variable		Overall (Number = 41)	Died (Number = 22)	Alive (Number = 19)	*p*-Value
AFP, ng/mL		22.6 (7.7–216.2)	46.2 (8.9–889.9)	13.7 (6.9–27.9)	0.060
AFP, ng/mL	<28.8	20 (57.1)	8 (40)	12 (80)	0.036
	≥28.8	15 (42.9)	12 (60)	3 (20)	
AST, U/L		58.5 (32.5–130.5)	80 (41–170)	37.5 (27–71)	0.030
ALT, U/L		43 (28–96)	49 (31–96)	32 (22–83)	0.167
Bilirubin, mg/dL		0.9 (0.57–2.07)	1.74 (0.8–2.97)	0.74 (0.47–0.9)	0.005
Albumin, g/L		40.85 (36.55–44.85)	38.6 (35.8–45)	42.55 (40.5–44.7)	0.175
CD4 cells count, mmc		368 (250–561)	305 (136–491)	400 (348–623)	0.035
CD4+/CD8 ratio		0.65 (0.38–1.01)	0.7 (0.38–1.15)	0.57 (0.37–0.96)	0.392
CD8 cells count, mmc		552 (379.5–915)	384.5 (272–562)	873 (513–1313)	0.0007
Neutrophils, 10^9^/L		3.8 (2.6–4.7)	3.3 (2.3–5.4)	3.9 (2.9–4.6)	0.570
Lymphocytes, 10^9^/L		1.35 (0.9–2.1)	0.95 (0.7–1.6)	1.95 (1.2–2.5)	0.005
Neutrophils/lymphocytes		2.46 (1.43–4.53)	3.5 (1.43–8.67)	2.06 (1.48–2.7)	0.068
Platelets, 10^9^/L		129.5 (83–189.5)	102.5 (66–165)	151.5 (124–204)	0.073
Creatinine, mg/dL		0.95 (0.78–1.2)	0.82 (0.76–0.98)	1.13 (0.94–1.44)	0.005
PTs/INR		1.07 (1–1.15)	1.15 (1.05–1.19)	1.02 (0.99–1.08)	0.008
HIV-RNA, copies/mL		377 (126–19621)	377 (140–12170)	13591 (111–27071)	0.999
HIV-RNA, ≥50 copies/mL		31 (75.6)	15 (68.2)	16 (84.2)	0.402
	Unknown	2 (4.9)	1 (4.5)	1 (5.3)	

Results are described by median (IQR) or frequency (%). Abbreviations: AFP: alpha phetoprotein; AST: aspartate aminotransferase (normal values < 35 U/L); ALT: alanine aminotransferase (normal values < 59 U/L); PTs/INR: prothrombin in second/international normalized ratio.

**Table 3 cancers-15-01653-t003:** Characteristics of PLWH with HCC at last available evaluation according to the primary study outcome (died or alive).

Variable		Overall (Number = 41)	Died (Number = 22)	Alive (Number = 19)	*p*-Value
CHILD Pugh Turcotte					0.563
	A	33 (80.5)	17 (77.3)	16 (84.2)	
	B	7 (17.1)	4 (18.2)	3 (15.8)	
	C	1 (2.4)	1 (4.5)	0 (0)	
BCLC					0.063
	0/A	20 (48.8)	7 (31.8)	13 (68.4)	
	B	8 (19.5)	6 (27.3)	2 (10.5)	
	C/D	13 (31.7)	9 (40.9)	4 (21.1)	
AFP, ng/mL		13.2 (2.7–333.2)	211 (20–3066′)	2.55 (2–6.0)	<0.0001
AFP, ng/mL					0.0002
	<28.8	24 (61.5%)	7 (33.3)	17 (94.4)	
	≥28.8	15 (38.5%)	14 (66.7)	1 (5.6)	
Albumin, g/L		36.69 (31.51–40.26)	33.1 (28.52–38.52)	40.27 (35.54–42.2)	0.002
AST, IU/L		58 (32–158)	126 (90–208)	34 (25–48)	0.0001
ALT, IU/L		43 (25–80)	63 (35–99)	28 (20–47)	0.008
Bilirubin, mg/dL		1.05 (0.67–1.94)	1.04 (0.53–1.78)	1.28 (0.7–2.09)	0.480
CD4 cell count, mmc		343 (215–497)	217.5 (143–341)	433 (348–614)	0.0004
CD4/CD8		0.7 (0.41–0.89)	0.64 (0.42–0.89)	0.72 (0.37–0.96)	0.855
CD8 cell count, mmc		546 (329–852)	383 (236–582)	848 (538–1139)	0.0005
Lymphocytes, 10^9^/L		1.5 (0.9–2)	0.9 (0.8–1.6)	1.9 (1.2–2.2)	0.002
Neutrophils, 10^9^/L		3.9 (2.9–5.4)	4.2 (2.3–6.8)	3.75 (3.1–4.1)	0.468
Neutrophils/lymphocytes		2.67 (2–4.2)	4.2 (2.67–5.53)	2.03 (1.59–2.91)	0.001
Platelets, 10^9^/L		151 (100–248)	110.5 (72–259)	170 (142–240)	0.187
Creatinine, mg/dL		1.02 (0.83–1.38)	0.93 (0.78–1.08)	1.14 (1.02–1.57)	0.023
PTs/INR		1.1 (1.03–1.19)	1.1 (1–1.22)	1.1 (1.03–1.18)	0.915
HIV-RNA, copies/mL		88 (64–12170)	65 (63–6118)	13591 (111–27071)	0.247
HIV-RNA, ≥50 copies/mL		6 (14.6%)	4 (18.2)	2 (10.5%)	0.489

Results are described by median (IQR) or frequency (%). Abbreviations: n: number; AFP: alpha phetoprotein; AST: aspartate aminotransferase (normal values < 35 IU/L); ALT: alanine aminotransferase (normal values < 59 IU/L); PTs/INR: prothrombin in seconds/international normalized ratio. BCLC: Barcelona Clinic Liver Cancer.

## Data Availability

All data generated or analyzed during this study are included in this article and its Appendix A. Further enquiries can be directed to the corresponding author.

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
