# Peer review of "Survival in People Living with HIV with or without Recurrence of Hepatocellular Carcinoma after Invasive Therapy"

_cancers, 2023, doi:10.3390/cancers15061653_

Round 1

Reviewer 1 Report

Dear all authors,

The entitled "survival in people living with HIV with or without recurrence of hepatocellular carcinoma after invasive therapy" provides very useful information to experts in the field as well as general readers. Personally, I am really interested to read your manuscript, too, and especially your suggested AFP cut-off levels. 

<Major>

1. It would be helpful to explain in a little detail about the markers of hepatocarcinoma such as CD4, and CD8 that you mentioned, in the introduction.  Because there are other biomarkers that have been suggested in many clinical studies. 

2. Is there any record of alcohol consumption or work status of participants? If there is, please put it into the tables.

3. I totally agreed with you in the discussion. The short number of participants is the weak point of the present study. However, based on the results of this study, it would be nice to supplement a little more advice on predictive and preventive medicine for patients and clinical experts in the conclusion.

<Minor>

The format of the tables should be improved.

English would be improved more.

References should be updated more. 

Author Response

  1. It would be helpful to explain in a little detail about the markers of hepatocarcinoma such as CD4, and CD8 that you mentioned, in the introduction.  Because there are other biomarkers that have been suggested in many clinical studies. 

Thank you for your comment. We added further information about hepatocellular carcinoma markers and specifically on CD4 in the section introduction (page 2), as suggested. References were added concerning this issue (references 15-21).

  1. Is there any record of alcohol consumption or work status of participants? If there is, please put it into the tables.

 Thank you for your comment. This is a very important issue, unfortunately this information was not routinely recorded in our database.

  1. I totally agreed with you in the discussion. The short number of participants is the weak point of the present study. However, based on the results of this study, it would be nice to supplement a little more advice on predictive and preventive medicine for patients and clinical experts in the conclusion.

Thank you for your comment. In our study AFP is the only predictive marker for survival. This finding is added in the section Conclusions. We could not give further advice to clinical experts because of the small sample size.  However, we suggested that “HCC/PLWH should have the same treatment opportunities as HIV negative participants also in term of re-treatment. We also suggested that  as for the counterpart of HIV uninfected participants, a more aggressive treatment than what were recommended by the BCLC system could be offered after careful selection to HCC/PLWH.”

Therefore we included in the section conclusion information on the HCC management in HCC/PLWH, while we did not explore specifically the aspect of preventive medicine.

<Minor>

The format of the tables should be improved.

Thank you for your comment. Even though our tables had the same format of the example in the journal’s template, we changed it with a new one that could make the reading easier for readers.

English would be improved more.

Thank you for your comment. We brought some changes in the language of our text.

References should be updated more. 

Thank you for your comment. We tried to update the references. We included three updated studies (references 15,26,29).

Reviewer 2 Report

Bertoni et al. reported that survival rates in PLWH with or without recurrence of HCC after invasive therapy. Please mention the definition of invasive therapy for HCC in details.

1.       In Material and methods section, “…and HIV parameters. AFP levels were”?

2.       In Tables 1, and 2, how many patients used NUC for their HBV infection? How many patients received DAAs for their HCV infection?

3.       In Discussion section, “However, more recent findings have shown that androgen receptor (AR) may inhibit the metastasis of late-stage HCC [31]. It was also reported that AR has an inhibitory role in the carcinogenesis of HCC by lowering the recurrence rate in post-surgery patients [32]. Although we did not investigate the androgen pathway, we can hypothesize that androgens can function in a contrasting manner and so can the estrogen pathway in the context of recurrence after invasive therapy.” Authors should discuss more this part and see the following references: Kanda T, et al. Androgen Receptor Could Be a Potential Therapeutic Target in Patients with Advanced Hepatocellular Carcinoma. Cancers (Basel). 2017 May 5;9(5):43. doi: 10.3390/cancers9050043. PMID: 28475115; Kanda T, Yokosuka O. The androgen receptor as an emerging target in hepatocellular carcinoma. J Hepatocell Carcinoma. 2015 Jun 26;2:91-9. doi: 10.2147/JHC.S48956. PMID: 27508198

Author Response

Bertoni et al. reported that survival rates in PLWH with or without recurrence of HCC after invasive therapy. Please mention the definition of invasive therapy for HCC in details.

Thank you for your comment. We detailed the definition of invasive therapy in the abstract section. It was also specified in the section Material and Method as following:  “This is a retrospective study on 41 adult PLWH with incident HCC diagnosed between November 2000 and October 2021, treated with IT defined as liver resection (LR) orthotopic liver transplantation (OLT) radiofrequency thermo-ablation (RFTA) trans arterial chemo or radioembolization (CRE).”

  1. In Material and methods section, “…and HIV parameters. AFP levels were”?

AFP levels were considered as median levels and were also stratified according to the cut-off levels of 28 ng/mL. This was clearly explained in the section Material and Methods. Otherwise, we kindly ask to rephrase the question as we couldn’t understand its meaning.

  1. In Tables 1, and 2, how many patients used NUC for their HBV infection? How many patients received DAAs for their HCV infection?

Thank you for your comment. We included data about treatment with NUC in HIV/HBV infected participants and HCV treatment with DAA  or interferon-ribavirin in HIV/HCV infected partecipants in the section Results.

  1. In Discussion section, “However, more recent findings have shown that androgen receptor (AR) may inhibit the metastasis of late-stage HCC [31]. It was also reported that AR has an inhibitory role in the carcinogenesis of HCC by lowering the recurrence rate in post-surgery patients [32]. Although we did not investigate the androgen pathway, we can hypothesize that androgens can function in a contrasting manner and so can the estrogen pathway in the context of recurrence after invasive therapy.” Authors should discuss more this part and see the following references: Kanda T, et al. Androgen Receptor Could Be a Potential Therapeutic Target in Patients with Advanced Hepatocellular Carcinoma. Cancers (Basel). 2017 May 5;9(5):43. doi: 10.3390/cancers9050043. PMID: 28475115; Kanda T, Yokosuka O. The androgen receptor as an emerging target in hepatocellular carcinoma. J Hepatocell Carcinoma. 2015 Jun 26;2:91-9. doi: 10.2147/JHC.S48956. PMID: 27508198

Thank you for your comment. We included a comment on the role of androgen and androgen receptors in hepatocarcinogenesis and associated references. However, we prefered not to discuss further this issue because, as indicated in the discussion section, we did not explore the significance of androgen pathway in promoting liver carcinogenesis and/or HCC recurrence. We simply found a lower prevalence of HCC recurrence among male participants.